# The Tree Autoencoder Model, with Application to Hierarchical Data Visualization

**Miguel Á. Carreira-Perpiñán**
Dept. of Computer Science and Engineering
University of California, Merced
mcarreira-perpinan@ucmerced.edu

**Kuat Gazizov**
Dept. of Computer Science and Engineering
University of California, Merced
kgazizov@ucmerced.edu

## Abstract

We propose a new model for dimensionality reduction, the PCA tree, which works like a regular autoencoder, having explicit projection and reconstruction mappings. The projection is effected by a sparse oblique tree, having hard, hyperplane splits using few features and linear leaves. The reconstruction mapping is a set of local linear mappings. Thus, rather than producing a global map as in t-SNE and other methods, which often leads to distortions, it produces a hierarchical set of local PCAs. The use of a sparse oblique tree and of PCA in its leaves makes the overall model interpretable and very fast to project or reconstruct new points. Joint optimization of all the parameters in the tree is a nonconvex nondifferentiable problem. We propose an algorithm that is guaranteed to decrease the error monotonically and which scales to large datasets without any approximation. In experiments, we show PCA trees are able to identify a wealth of low-dimensional and cluster structure in image and document datasets.

## 1 Introduction

As a form of exploratory data analysis, dimensionality reduction (DR) for visualization seeks to provide a representation of a dataset in dimension at most 3 (we will focus on 2D) that, combined with the human visual ability to pick patterns, is able to find structure of interest in a high-dimensional dataset. Many DR methods have been proposed over the years. Currently, $t$-SNE [40] and PCA are the most popular choices and have seen widespread application in many areas.

Here we advocate for a different approach, that of hierarchical local linear projections, which we call the **PCA tree**. Hierarchical DR is not new, but what is new is our definition of the model as a certain type of tree autoencoder and our algorithm to learn it by minimizing the reconstruction error in a self-supervised way, which brings considerable advantages in interpretability and training and inference time. From the outset, we acknowledge that DR is a hard, ill-defined problem that no single technique can solve satisfactorily. In reducing a dataset to 2D we necessarily have to lose information and produce artifacts. Indeed, the distortions of $t$-SNE and their impact on applications have been well documented [6, 22]. However, the PCA tree provides significant, complementary advantages over previous methods.

Firstly, *PCA trees are practical and intuitive to use*. They optimize the reconstruction error, which has a clear meaning. They do not require a neighborhood graph (and perplexity parameter, etc.), which is tricky to estimate so it captures manifold structure, and computationally very costly. The model, a **tree autoencoder** (see later), *directly defines nonlinear out-of-sample mappings (encoder and decoder)*. Its hyperparameters are natural: the *depth* $\Delta$ of the tree, which directly controls the resolution (number of scatterplots and reconstruction error), and the *sparsity* $\lambda$, which helps to interpret the tree decision nodes. Inference on a tree (encoding or decoding) is extremely fast. Training

38th Conference on Neural Information Processing Systems (NeurIPS 2024).

(to find a local optimum, a hard problem because the tree is nondifferentiable) is linear on the sample size and parallelizable, hence *scalable to large datasets without the need for approximations*.

Second, *PCA trees are highly interpretable* and, as demonstrated in our experiments, they extract a wealth of information from complex datasets. This comes from the fact that both trees and PCA are techniques with well understood interpretability. Specifically: 1) the use of trees to construct a nested hierarchy of maps where an input instance follows a single root-leaf path; 2) the use of sparse oblique trees, with hyperplanes using few features at the decision nodes; 3) the use of linear projections in the leaves, each a PCA scatterplot; and 4) the ability to see the data at multiple scales.

Third, *a PCA tree uses multiple local latent projections (an "album") hierarchically organized rather than a single global one ("atlas")*. The latter is more likely to introduce distortions in trying to arrange many dimensions into a 2D map, and is hard to use with large datasets, requiring a lot of pan and zoom. This is less of a problem if one focuses on a local part of the data, particularly if there are clusters. Also, a linear projection is easy to understand and, while it does introduce distortions, it is well understood how this happens: distances can only shrink. If a PCA map shows clusters, these are real—in contrast with $t$-SNE's tendency to create false clusters [6, 22], for example. Also, PC directions often correspond to some real degree of freedom, and the mean is obviously a useful summary of the data. Thus, while global PCA generally collapses different parts of a manifold, local PCAs can succeed—if properly learned. Multiple local maps are like an album of a house, where each picture shows a different room, but with an hierarchical structure.

A final advantage is that the loss function is really a self-supervised regression problem. This makes it *possible to use cross-validation to determine the hyperparameters* (although, in an exploratory data analysis, we probably want to explore such hyperparameters manually).

The PCA tree does have its limitations, as follows. It needs access to the explicit feature vectors, unlike graph-based methods, which need instead pairwise affinities. The optimization converges to a local optimum, which depends on the initialization. And, as with any DR method, PCA trees reveal only some structure, so practitioners should try multiple, complementary methods.

## 2 Related work

The literature on DR is vast. We note connections to the most relevant methods.

**Non-hierarchical DR** Of the many existing DR methods, we mention two large groups that are particularly important in this paper. One are *autoencoders*, which learn an encoder or projection mapping and a decoder or reconstruction mapping so as to minimize the reconstruction error of each training point. The classic examples are PCA and neural network autoencoders [34, 17], where the encoder and decoder are linear and nonlinear, respectively. Our tree autoencoders are another example. The other are *nonlinear embedding methods*, which learn the 2D projection of each training point so that Euclidean distances in latent space best match distances or similarities in the high-dimensional space. These include multidimensional scaling [5], SNE [16], $t$-SNE [40], the elastic embedding [6] and UMAP [28], among others. Although often proposed as matching distributions of neighbors in the high- and low-dimensional spaces, these methods can be equivalently seen as optimizing a tradeoff of attraction between true neighbors and repulsion between points [6, 41, 22].

**Hierarchical DR** A multiscale view of DR is an attractive idea that has been explored before, often justified by the fact that a single 2D map may miss much of the structure in a complex dataset. We consider two ways of defining the hierarchical structure: soft and hard. In *soft hierarchies*, one defines a soft, probabilistic tree [20] so that an input instance traverses every tree path with a positive probability. Thus, the model really is a probabilistic mixture of the leaves ("experts") with mixing proportions given by the tree structure. This has the advantage that the model is differentiable, so optimization is straightforward with gradient- or EM-based approaches. Unfortunately, we lose the interpretability and fast training and inference, since an instance reaches all nodes, and the leaves may not localize in input space and overlap with each other. Examples of this are [4, 30, 38, 19]. A more detailed comparison of soft and hard oblique decision trees appears in [12].

Making the tree decisions hard makes the hierarchy nested, but the optimization is much more difficult (indeed, this is an important contribution in our paper). One can construct a hard hierarchy in a greedy sequential way by recursively partitioning the dataset, or by using some other tree con-

struction (such as kd-trees [11] or cover trees [2]); and then applying some form of DR at each node. However, this is suboptimal (since the node parameters are fixed as they are added, or independently from the leaf parameters) and produces overly large trees. Conversely, one can grow the tree bottom-up by a form of hierarchical clustering, but this is also suboptimal. These approaches include work such as [1, 24, 26, 29] and others such as hierarchical self-organizing maps [32], sequential NMF [9] and hierarchical SNE [31].

**Flat, local DR**  This seeks to estimate a collection of local latent maps but without any hierarchy. Again, this can be done in a soft way via probabilistic mixtures (of factor analyzers [13], PCAs [37], etc.) or in a hard way as a form of $k$-PCAs [21]. A few works have also sought to align a collection of local DR models into a single global map [33, 36, 44, 23].

**Other related models**  Some works have tried to combine graph-based methods, such as $t$-SNE, with hierarchical DR. For any DR method that does not define an out-of-sample mapping, such as $t$-SNE and other nonlinear embedding methods, it is possible to learn a tree-based projection mapping [45]. Also, some fast approximations for graph-based embedding methods are based on tree data structures, such as Barnes-Hut trees [43, 39, 42, 35]. However, this is intended to accelerate the N-body computations, not to define a hierarchical DR.

In view of this, the main novelty of our work is in defining an autoencoder consisting of a hard tree encoder and a local linear decoder, and in giving an algorithm to learn it by optimizing the reconstruction error. We believe this is the first algorithm to guarantee a monotonic decrease of the error jointly over the tree parameters and local PCAs. We describe both the tree autoencoder model and the optimization algorithm next.

## 3    Definition of the PCA tree as an autoencoder

Like any autoencoder, the PCA tree defines an encoder $\mathbf{F}$ and decoder $\mathbf{f}$, but in a peculiar way, as follows. Firstly, consider a fixed rooted directed tree structure with decision nodes and leaves indexed by sets $\mathcal{D}$ and $\mathcal{L}$, respectively, and $\mathcal{N} = \mathcal{D} \cup \mathcal{L}$. Both the encoder and autoencoder use this tree structure. Each decision node $i \in \mathcal{D}$ has a decision function $h_i(\mathbf{x}; \mathbf{w}_i, w_{i0}) \colon \mathbb{R}^D \to \mathcal{C}_i$, where $\mathcal{C}_i = \{\texttt{left}_i, \texttt{right}_i\} \subset \mathcal{N}$, sending instance $\mathbf{x}$ to the corresponding child of $i$. Rather than axis-aligned trees, which are a poor model for high-dimensional data, we use oblique trees, having hyperplane decision functions "go to right if $\mathbf{w}_i^T \mathbf{x} + w_{i0} \geq 0$", with $\mathbf{w}_i \in \mathbb{R}^D$ and $w_{i0} \in \mathbb{R}$. The tree's prediction for an instance $\mathbf{x}$ is obtained by routing $\mathbf{x}$ from the root to exactly one leaf and applying this leaf's predictor (defined below). We define the *reduced set (RS)* $\mathcal{R}_i \subset \{1, \ldots, N\}$ of a node $i \in \mathcal{N}$ as the training instances that reach $i$ given the current tree parameters.

**Encoder or projection mapping F**  This is given by a tree mapping $\mathbf{T}^e(\mathbf{x}; \boldsymbol{\Theta}) \colon \mathbb{R}^D \to \mathcal{L} \times \mathbb{R}^L$ where the predictor for leaf $j$ has the form of a linear mapping $\mathbf{F}_j(\mathbf{x}; \mathbf{U}_j, \boldsymbol{\mu}_j) = \mathbf{U}_j^T(\mathbf{x} - \boldsymbol{\mu}_j)$, where $\mathbf{U}_j \in \mathbb{R}^{D \times L}$ is an orthogonal matrix and $\boldsymbol{\mu}_j \in \mathbb{R}^D$. The encoder parameters are $\boldsymbol{\Theta} = \{\mathbf{w}_i, w_{i0}\}_{i \in \mathcal{D}} \cup \{\mathbf{U}_j, \boldsymbol{\mu}_j\}_{j \in \mathcal{L}}$. Thus, the encoder maps an input instance $\mathbf{x} \in \mathbb{R}^D$ to a leaf index $j \in \mathcal{L}$ and an $L$-dimensional real vector $\mathbf{z} = \mathbf{U}_j^T(\mathbf{x} - \boldsymbol{\mu}_j)$, which at an optimum (see later) will be the PCA projection in that leaf. This means that the PCA tree does not have a common latent space of dimension $L$ where all instances are projected. Instead, it has one separate $L$-dimensional PCA space per leaf. Thus, *the PCA tree latent space $\mathcal{L} \times \mathbb{R}^L$ is mixed discrete-continuous*[1], although it can also be seen as a separate continuous $L$-dimensional PCA space per leaf.

**Decoder or reconstruction mapping f**  This maps a leaf index $j$ and $L$-dimensional vector $\mathbf{z}$ (in $\mathcal{L} \times \mathbb{R}^L$) to a vector in $\mathbb{R}^D$. It consists of a set of linear mappings of the form $\mathbf{f}_j(\mathbf{z}; \mathbf{U}_j, \boldsymbol{\mu}_j) = \mathbf{U}_j \mathbf{z} + \boldsymbol{\mu}_j$ for $j \in \mathcal{L}$.

**Tree autoencoder**  We can now define the autoencoder as the composition of the decoder and encoder, $\mathbf{T} = \mathbf{f} \circ \mathbf{F}$. Conveniently, we can absorb the leaf index implicitly into each leaf. Thus,

---

[1]We could define the autoencoder so it uses a single, global latent space. However, this complicates the optimization (although it could be done with the method of auxiliary coordinates (MAC) of [8] by considering the problem as a nested function). Also, the per-leaf space arises naturally from the fact that the encoder partitions the space into leaf regions.

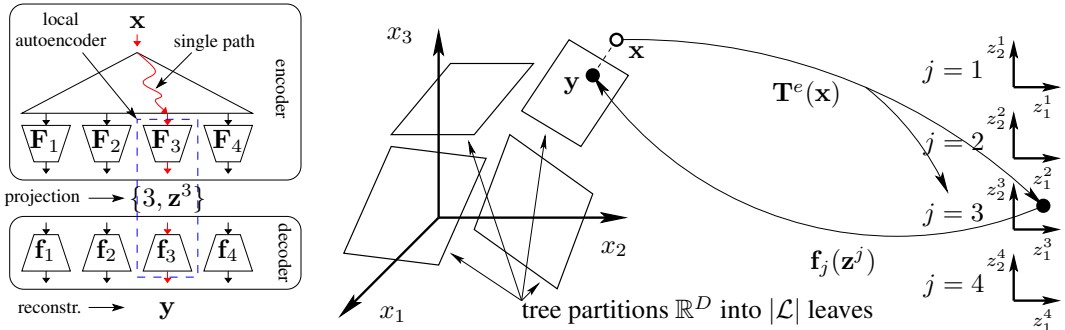

Figure 1: Illustration of the tree autoencoder model with a data space $\mathbb{R}^3$ for a tree of depth $\Delta = 2$ with leaves $\mathcal{L} = \{1, 2, 3, 4\}$, defining a latent space (album) of four 2D spaces.

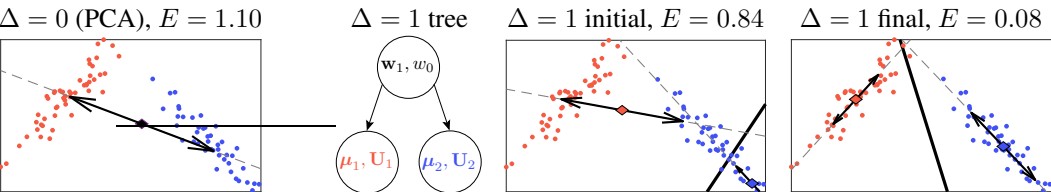

Figure 2: PCA tree on a 2D toy data set. *Left plot*: regular PCA (PCA tree with $\Delta = 0$). *Rest of plots*: PCA tree with $\Delta = 1$. Mean $\boldsymbol{\mu}_j$: diamond; principal component direction $\mathbf{U}_j$ and variance: dashed line and arrows; decision boundary $\mathbf{w}_1, w_{10}$ at the root: solid line.

$\mathbf{T}(\mathbf{x}; \boldsymbol{\Theta})$ is simply a regression tree identical to the tree encoder $\mathbf{T}^e(\mathbf{x}; \boldsymbol{\Theta})$ but where the predictor mapping of leaf $j$ has the form $\mathbf{g}_j(\mathbf{x}; \mathbf{U}_j, \boldsymbol{\mu}_j) = \mathbf{U}_j \mathbf{U}_j^T (\mathbf{x} - \boldsymbol{\mu}_j) + \boldsymbol{\mu}_j$. Note this is a linear mapping of rank $L < D$, and it is the composition $\mathbf{f}_j \circ \mathbf{F}$ of leaf $j$'s decoder with the encoder.

Fig. 1 illustrates the tree autoencoder idea. Fig. 2 shows a PCA tree on a 2D toy example using a latent dimension $L = 1$.

## 4 Optimization algorithm

Our objective function is now the regular reconstruction error of an autoencoder $\mathbf{T} \colon \mathbb{R}^D \to \mathbb{R}^D$ with an $\ell_1$ regularization term of hyperparameter $\lambda \geq 0$ on a training set $\{\mathbf{x}_n\}_{n=1}^N \subset \mathbb{R}^D$:

$$E(\boldsymbol{\Theta}) = \sum_{n=1}^N \|\mathbf{x}_n - \mathbf{T}(\mathbf{x}_n; \boldsymbol{\Theta})\|_2^2 + \lambda \sum_{i \in \mathcal{D}} \|\mathbf{w}_i\|_1 \quad \text{s.t.} \quad \mathbf{U}_j^T \mathbf{U}_j = \mathbf{I}, \ \forall j \in \mathcal{L}. \tag{1}$$

Thus, the PCA tree is trained like a regression tree which maps real vectors to real vectors. It consists of a fixed tree structure (which we will take as complete of depth $\Delta$); sparse oblique decision nodes, consisting of a hyperplane split using few features (as encouraged by the $\ell_1$ sparsity penalty); and linear low-rank leaves. The following theorem explains why we call it "PCA tree" (in fact, we designed the form of the loss function and tree so that this would happen).

**Theorem 4.1** (optimality condition over the leaves). *The optimal $\{\mathbf{U}_j^*, \boldsymbol{\mu}_j^*\}_{i \in \mathcal{L}}$ given the remaining parameters are fixed, i.e.:*

$$\min_{\{\mathbf{U}_j, \boldsymbol{\mu}_j\}_{i \in \mathcal{L}}} E(\boldsymbol{\Theta}) \quad s.t. \quad \mathbf{U}_j^T \mathbf{U}_j = \mathbf{I}, \ \forall j \in \mathcal{L} \tag{2}$$

*corresponds to setting each $\{\mathbf{U}_j^*, \boldsymbol{\mu}_j^*\}$ to the $L$ principal components of $\mathcal{R}_j$, the RS of leaf $j$.*

*Proof.* Because of the separability condition (theorem 4.2) and the fact that $\{\mathbf{w}_i, w_{i0}\}_{i \in \mathcal{D}}$ are fixed, problem (2) is equivalent to

$$\min_{\{\mathbf{U}_j, \boldsymbol{\mu}_j\}_{i \in \mathcal{L}}} \sum_{n \in \mathcal{R}_j} \left\|\mathbf{x}_n - \mathbf{g}_j(\mathbf{x}_n; \mathbf{U}_j, \boldsymbol{\mu}_j)\right\|_2^2 \quad \text{s.t.} \quad \mathbf{U}_j^T \mathbf{U}_j = \mathbf{I}, \ \forall j \in \mathcal{L} \tag{3}$$

which (since the objective and constraints separate) is also equivalent to solving this for each leaf $j \in \mathcal{L}$ separately:

$$\min_{\mathbf{U}_j, \boldsymbol{\mu}_j} \sum_{n \in \mathcal{R}_j} \left\| \mathbf{x}_n - (\mathbf{U}_j \mathbf{U}_j^T (\mathbf{x} - \boldsymbol{\mu}_j) + \boldsymbol{\mu}_j) \right\|_2^2 \quad \text{s.t.} \quad \mathbf{U}_j^T \mathbf{U}_j = \mathbf{I}. \tag{4}$$

The solution of the latter problem is well known to be given by PCA, i.e., $\boldsymbol{\mu}_j^* = \frac{1}{|\mathcal{R}_j|} \sum_{n \in \mathcal{R}_j} \mathbf{x}_n$ is the mean of the training instances in $\mathcal{R}_j$ and $\mathbf{U}_j^*$ consists of the $L$ eigenvectors of the covariance matrix $\boldsymbol{\Sigma}_j = \frac{1}{|\mathcal{R}_j|} \sum_{n \in \mathcal{R}_j} (\mathbf{x}_n - \boldsymbol{\mu}_j^*)(\mathbf{x}_n - \boldsymbol{\mu}_j^*)^T$ associated with its largest $L$ eigenvalues. $\qquad \square$

We borrow the idea of alternating optimization over the tree nodes from [7] (Tree Alternating Optimization (TAO)), which is fundamental to be able to update the parameters so as to reduce the objective function. Firstly, at any decision tree making hard decisions, each input instance follows exactly one root-leaf path. This results in the following *separability condition*.

**Theorem 4.2** (Separability condition). *Let $\mathbf{T}(\mathbf{x}; \boldsymbol{\Theta})$ be the autoencoder tree and $\mathcal{S} \subset \mathcal{N}$ a nonempty set of nodes that are not descendant from each other. Call $\boldsymbol{\theta}_i$ the parameters in node $i$. Then, as a function of the parameters $\{\boldsymbol{\theta}_i \colon i \in \mathcal{S}\}$ (i.e., fixing all other parameters $\boldsymbol{\Theta}_{rest} = \boldsymbol{\Theta} \setminus \{\boldsymbol{\theta}_i \colon i \in \mathcal{S}\}$), the function $E(\boldsymbol{\Theta})$ of eq. (1) can be equivalently written as*

$$E(\boldsymbol{\Theta}) = \sum_{i \in \mathcal{S}} E_i(\boldsymbol{\theta}_i, \boldsymbol{\Theta}_{rest}) + E_{rest}(\boldsymbol{\Theta}_{rest}) \tag{5}$$

*where $\{E_i \colon i \in \mathcal{S}\}$ and $E_{rest}$ are certain functions.*

*Proof.* Analogous to that in [7]. It follows from the fact that the reduced sets of all nodes in $\mathcal{S}$ are disjoint, because the tree makes hard decisions. $\qquad \square$

That is, optimizing over the parameters of any set of nodes which are not descendants of each other separates: we can equivalently optimize each node on its own over its parameters and using only its RS. This simplifies the problem considerably and introduces significant parallelism: for example, all nodes at the same depth can be optimized in parallel. Indeed, our overall algorithm will process nodes from the leaves towards the root, optimizing all nodes at the same depth at each step.

Second, we still have to solve the problem of optimizing (1) over the parameters of one given node. This simplifies into a *reduced problem (RP)*. If the node is a *leaf* $j$, theorem 4.1 gives us the answer: we set the leaf's $\mathbf{U}_j, \boldsymbol{\mu}_j$ to the PCA on $\mathcal{R}_j$. This is the exact solution to the optimization over the leaves, so the overall objective function value will either decrease or stay constant. If the node is a *decision node* $i$, optimizing (1) over $\mathbf{w}_i, w_{i0}$ is equivalent to a RP as given by the following theorem.

**Theorem 4.3** (Reduced problem over a decision node). *Consider the objective function $E(\boldsymbol{\Theta})$ of eq. (1) and a decision node $i \in \mathcal{D}$. Assume the parameter values of all the nodes except $i$ are fixed. Then, the optimization problem $\min_{\mathbf{w}_i, w_{i0}} E(\boldsymbol{\Theta})$ is equivalent to the following problem:*

$$\min_{\mathbf{w}_i, w_{i0}} \overline{E}_i(\mathbf{w}_i, w_{i0}) = \sum_{n \in \mathcal{R}_i} \overline{L}_{in}(\overline{y}_{in}, h_i(\mathbf{x}_n; \mathbf{w}_i, w_{i0})) + \lambda \|\mathbf{w}_i\|_1 \tag{6}$$

*where: $\mathcal{R}_i$ is the reduced set of node $i$; we define the weighted 0/1 loss $\overline{L}_{in}(\overline{y}_{in}, \cdot) \colon \mathcal{C}_i \to \mathbb{R}^+ \cup \{0\}$ for instance $n \in \mathcal{R}_i$ as $\overline{L}_{in}(\overline{y}_{in}, y) = l_{in}(y) - l_{in}(\overline{y}_{in}) \; \forall y \in \mathcal{C}_i$; we define the pseudolabel $\overline{y}_{in} = \arg\min_{y \in \mathcal{C}_i} l_{in}(y)$ as the "best" child of $i$ for $n$ (or any $\overline{y}_{in} \in \arg\min_{y \in \mathcal{C}_i} l_{in}(y)$ in case of ties); and we define the function $l_{in} \colon \mathcal{C}_i \to \mathbb{R}$ as $l_{in}(z) = \|\mathbf{x}_n - \mathbf{T}_z(\mathbf{x}_n; \boldsymbol{\Theta}_z)\|_2^2$ for any $z \in \mathcal{C}_i$ (child of $i$), where $\mathbf{T}_z(\cdot; \boldsymbol{\Theta}_z)$ is the predictive function for the subtree rooted at node $z$.*

*Proof.* Analogous to that for the decision node RP in [7]. A difference is that there the tree was a classification tree with the 0/1 loss, which resulted in an unweighted problem; while here we use the reconstruction error, which results in a weighted problem in (6). This follows from the fact that all a decision node can do with an instance is send it down its left or right child, and the ideal choice is the one that results in the lowest reconstruction error downstream from that node. $\qquad \square$

The *reduced problem* of eq. (6) is an $\ell_1$-*regularized weighted 0/1 loss binary classification problem* over a linear classifier $h_i$ with binary *pseudolabels* $\{\overline{y}_{in}\}$, defined as the child of $i$ that gives the best reconstruction for $\mathbf{x}_n$ under the current tree. This problem is NP-hard, but we can typically

get a good approximate solution by solving an $\ell_1$-regularized surrogate loss instead. We use the logistic loss with instances weighted as in theorem 6. We can guarantee that the overall objective function value decreases or stays constant by accepting the surrogate solution only if it improves over the previous parameters[2], which empirically is usually the case. This means each node update, hence each overall iteration over all nodes, monotonically decreases (1). We stop when there is little change in the parameters or objective, or when we reach a set number of iterations.

This concludes the algorithm, whose pseudocode is in fig. 6. Essentially, it repeatedly updates the nodes in turn: at a leaf it solves a PCA, and at a decision node it solves an $\ell_1$-regularized, instance-weighted logistic regression problem (we use LIBLINEAR [10]). The tree is initialized as a random median tree, i.e., at the root we pick a random direction for the weight vector and a bias so that half of the instances go to either child; and we repeat this recursively down the tree.

## 5 Computational complexity for training and inference

Throughout, assume a complete tree of depth $\Delta$; sparse decision weight vectors having $sD$ nonzero values on average, with $s \in [0, 1]$ (so $s = 1$ means dense vectors); and a latent dimension $1 \leq L \leq D$ at each leaf ($L \leq 3$ for visualization). As cost we use the number of scalar multiplications.

### 5.1 Inference

By inference we mean the time to map an input instance $\mathbf{x} \in \mathbb{R}^D$ to its leaf $j \in \mathcal{L}$ and latent vector $\mathbf{z} \in \mathbb{R}^L$. This requires traversing a single root-leaf path of depth $\Delta$, at a cost $\Theta(sD\Delta)$, and computing the leaf PCA projection, at a cost $\Theta(LD)$, total $\Theta((L + s\Delta)D)$ (which, for $\Delta = 0$, i.e., regular PCA, corresponds to $\Theta(LD)$). This is very fast, since both $L$ and $\Delta$ are very small. Reconstructing a latent vector $\mathbf{z} \in \mathbb{R}^L$ at a leaf $j \in \mathcal{L}$ costs $\Theta(LD)$.

### 5.2 Training

The cost of one iteration, i.e., updating each node in the tree once, is as follows. *For the decision nodes*, this equals to setting up the RP (i.e., computing the pseudolabel losses) and solving it with one logistic regression in each node's RS. To work this out we note that the RSs of all nodes at the same depth total $N$ instances, and we assume logistic regression to be linear in the dimensionality and sample size. Then, the total cost of all decision nodes has two terms. The first is the pseudolabel cost over all decision nodes, which is $\Theta(\frac{1}{2}sND\Delta(\Delta - 1))$. This is due to the fact that, at each decision node, we have to send each training instance down the left and right children to compute their losses (actually, we need only send it down one child if we precompute the losses), and the decision nodes span depths from 0 to $\Delta - 1$. The second is the RP solution, whose cost is proportional to that of running $\Delta$ logistic regressions on the whole dataset (at depths $0, 1, \ldots, \Delta - 1$), or $\Theta(c_1 ND\Delta)$, where $c_1 > 0$ is a constant factor (which includes the average number of iterations of the logistic regression solver). This cost is as in the sparse oblique classification trees of [7]. Asymptotically, it totals $\Theta(ND\Delta^2)$.

*For the leaves*, we have a peculiar situation. Say a leaf has a RS with $M$ instances. The cost of PCA[3] on that RS is $\Theta(MD^2 + c_2 D^3)$ if $D \leq M$ and $\Theta(DM^2 + c_2 M^3)$ if $D \geq M$, depending on whether we compute the eigenvectors of the $D \times D$ covariance matrix or of the $N \times N$ Gram matrix, respectively, where $c_2 > 0$ is a constant factor. Assume for simplicity that each RS has $M = N2^{-\Delta}$ instances (balanced partition) and define the *critical depth* $\Delta^* = \log_2\left(\frac{N}{D}\right)$. Then:

$$\begin{matrix} \text{cost, all} \\ \text{leaves} \end{matrix} = \begin{cases} \Delta \leq \Delta^*, \text{ shallow tree regime:} & \Theta(ND^2 + c_2 D^3 2^\Delta) \\ \Delta \geq \Delta^*, \text{ deep tree regime:} & \Theta(N^2 D 2^{-\Delta} + c_2 N^3 2^{-2\Delta}) \end{cases} \leq \Theta((1 + c_2)ND^2).$$

For a given dataset of fixed $N$ and $D$, the cost depends on the tree depth $\Delta$ and has a turning point when $\Delta = \Delta^*$. For datasets having[4] $N < D$, we are always in the deep regime, but if $N > D$ then we can be in the shallow or deep regime for small or large $\Delta$, respectively. Interestingly, this leads

---

[2]In fact, in practice we find that always accepting the update (even if it increases the objective) works as well and is simpler. This makes the iterates slightly noisy, but over iterations the objective decreases steadily.

[3]It is possible to accelerate this via approximate SVD algorithms or matrix sketches to compute just the top eigenvectors, as well as using a good initialization, but here we consider the cost of a full eigendecomposition.

[4]If $N < D$ we can always run PCA first and reduce the data to $D < N$ without error.

to a non-monotonic cost as a function of $\Delta$, as shown in fig. 3, because the shallow regime cost increases with $\Delta$ but the deep regime one decreases with $\Delta$. This is surprising because the deeper the tree the more leaves it has, yet the faster they train. The reason is that the PCA cost is superlinear on the sample size, but the sample size per leaf decreases proportionally to the number of leaves.

We emphasize that, as noted above, in both regimes the leaves' cost is strictly upper bounded by $(1 + c_2)ND^2$ (with equality if $\Delta = \Delta^*$), hence *this cost is at most linear in $N$ and quadratic in $D$, just like in regular PCA*. For fixed $N$ and $D$, the leaves cost first dominates the decision nodes cost (shallow tree regime) because the former includes a term $ND^2$ that dominates the decision nodes' term $\Theta(ND\Delta^2)$, since $\Delta^2 < D$ in practical cases. But as $\Delta$ grows (deep tree regime) the leaves cost decreases and reaches $N(D + c_2)$ for $\Delta = \log_2 N$ (when each leaf contains a single instance), so the overall cost is eventually dominated by the decision nodes, $\Theta(ND\Delta^2)$. As a useful summary, for fixed $\Delta$ *the rough cost is $\Theta(ND^2)$ for shallow trees and $\Theta(ND)$ for deep trees*— which is asymptotically faster than PCA! (although very deep trees having few instances per leaf are likely not practical). This makes the algorithm highly scalable to large sample sizes without the need for any approximation. This is a significant advantage over neighbor embedding algorithms such as $t$-SNE, which have a quadratic cost on $N$ and require some approximation to reduce this [39, 42, 43].

**Parallel training**   From the separability condition, all nodes at the same depth can be updated in parallel, which accelerates the training considerably. Assume for simplicity we have $2^\Delta$ processors (which is perfectly feasible for trees having depth 4, as in our experiments). Then the costs are as follows. For decision nodes, we have the cost of running a logistic regression on $\Delta$ datasets of sizes $N, \frac{N}{2}, \ldots, N2^{-\Delta}$ (from the root to the leaf parents), a geometric series which is upper bounded by $2N$, hence a total cost of $\Theta(2c_1 ND)$, i.e., $\frac{\Delta}{2}$ *times faster than the sequential computation*. (The pseudolabel computation separates over instances and is thus $2^\Delta$ times faster.) For the leaves, we have the cost of running PCA in a single leaf, i.e., $2^\Delta$ *times faster than the sequential computation*. Again, this is a significant advantage over $t$-SNE, which cannot be easily parallelized.

# 6   Experiments

In summary, we show the following: we confirm our theoretical predictions about monotonic decrease of the objective function and training time; compare the reconstruction error with PCA; and demonstrate how PCA trees are highly interpretable and extract significant structure from complex datasets. We use several datasets of different types (images, documents), some of which appear in the appendix.

## 6.1   Training time and scalability to large datasets

Fig. 3 (right 2 plots) shows learning curves for different datasets and tree depths. As can be seen, the objective function decreases monotonically over iterations, and converges to a near-optimal solution in usually less than 10 iterations.

Fig. 3 (left 2 plots) shows the training time per iteration in seconds (theoretical and actually measured for the MNIST dataset), as a function of the tree depth $\Delta$ in one processor (sequential computation, without parallelism). For the theoretical estimate we used constant factors $c_1 = 1$, $c_2 = 4$ (see section 5) and $s = 1$ (dense decision node weight vectors). To vary the dimensionality $D \in \{64, 784, 1600, 2500\}$, we subsampled or oversampled the MNIST images to $8 \times 8$, $28 \times 28$ (original size), $40 \times 40$ and $50 \times 50$ pixels, respectively. The plots confirm what we described in our complexity analysis: the computation at the leaves first increases until $\Delta = \Delta^* = \log_2\left(\frac{N}{D}\right)$, then decreases, resulting in a non-monotonic total training time. The critical depth $\Delta^*$ at which the change happens decreases with $D$, from around 10 to 5; note the actual depth at which the total time peaks is different, because the latter also considers the decision nodes' cost, which is linear on $\Delta$.

Fig. 3 (plot 3) shows scalability to larger sample sizes using the Infinite MNIST dataset [27] (images of $28 \times 28$, so $D = 784$). We ran the PCA tree for 10 iterations, when it approximately converged. The linear cost of the PCA tree is clear as a slope 1 in the log-log plot. Training on 1M samples with depth 4 takes less than an hour. In contrast, $t$-SNE [39] and UMAP [28] are slower and have a superlinear cost, even though the implementations we used are not exact (they use approximations, such as Barnes-Hut trees or neighbor subsampling, to accelerate the computations, besides finding approximate rather than nearest neighbors).

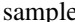
sample

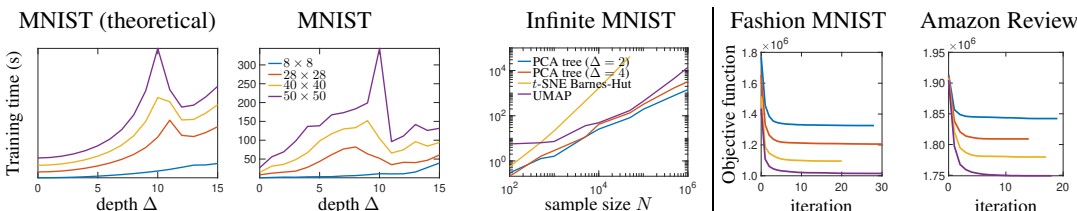

Figure 3: *Plots 1, 2*: training time per iteration on (Infinite) MNIST for PCA trees with different $N$, $D$, $\Delta$. *Plot 3*: training time for PCA trees (10 iterations) and for $t$-SNE and UMAP (default user parameters). *Plots 4, 5*: learning curves for PCA trees of different depths on several datasets.

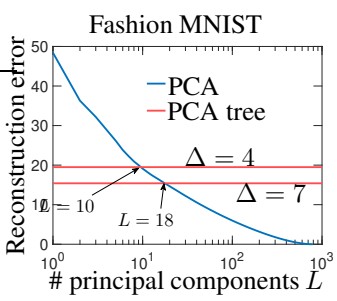
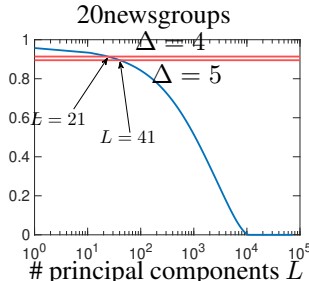

Figure 4: Squared reconstruction error per sample of PCA as a function of the number of principal components $L$ and of PCA tree using $L = 2$ PCs in each leaf, for different depths on two datasets: Fashion MNIST ($N = 60000$, $D = 784$) and 20newsgroups ($N = 11314$, $D = 72764$).

## 6.2 Reconstruction error

Fig. 4 shows the reconstruction error for PCA and PCA tree. Although the PCA tree uses $L = 2$ in each leaf, it is able to achieve a much lower error than PCA, which would need a much larger number of components to match it. This is the result of the PCA tree ability to learn a partition of the data and local PCAs that are jointly optimal.

## 6.3 Interpreting and exploring PCA tree visualizations

Fig. 5 (see also figs. 8–10 in the appendix) shows results for the Fashion MNIST dataset[5]. This consists of $N = 60\,000$ grayscale images of $28 \times 28$ pixels (so $D = 784$), each showing a clothing or shoe item, labeled into 10 balanced classes. The features are the pixel intensities in [0,1]. The class labels were not used during training, they were only used for visualization purposes. We show a PCA tree of depth 4 (using $\lambda = 100$), thus having 15 decision nodes (with weight vectors shown as $28 \times 28$ images) and 16 leaves (showing a 2D PCA scatterplot and the mean and PCs as images). The appendix contains results for other datasets (images: MNIST, documents: Amazon Reviews). Even at a glance we can see lots of structure in the tree: many classes organize hierarchically in a meaningful way; decision weight vectors are sparse and highly meaningful, often separating groups of classes by focusing on the discriminant regions of the image; the PCs in the leaves usually are highly meaningful, indicating the change of some relevant degree of freedom; and the leaf scatterplots show clusters or class structure; among other things. We comment on some of these in detail. For reference, fig. 9 (appendix) shows the 2D global plot by PCA, $t$-SNE and other algorithms.

**Global tree structure** Although our algorithm is unsupervised, it has the ability to separate different types of classes. For example, leaves L1 and L2 contain different types of shoes, while leaf L14 contains only bags. Some classes (e.g. Pullover and Shirt in L8) differ in tiny details and can only be separated if using the labels (remember that minimizing the reconstruction error seeks maximum-variance directions, which are not necessarily aligned with class variation). The coordinate axes in the leaves show the most variance in the data within the leaf's region. Since the RSs are nested along the tree hierarchy, we can also show a scatterplot at each decision node (appendix fig. 9), which shows how classes and other patterns are progressively separated as one descends deeper into the tree. However, note the optimal parameters if the tree was shallower would differ from those.

**Local tree structure** The decision nodes' weight vectors select features sparsely, often focusing on parts of the image that can differentiate groups of images and send them selectively to the left or

---

[5]It is instructive to compare these results with those of training a sparse oblique *classification* tree on the Fashion MNIST dataset [14], as there are interesting similarities and differences.

right subtree. The figure is annotated in some places to make this clear. For example, decision node D2 separates clothes "with" and "without" sleeves. In some cases, this results in the tree separating actual classes. For instance, node D14 separates "outerwear" type of clothes from "bags", which is explainable by simply looking at the decision node weights and the part of the image where they focus: blue values indicate "if the image has a neckline and shoulders" (like outwear clothes) and red values indicate "if it has wide frames" (like bags).

**PCA at the leaves** Since each leaf applies 2D PCA on its corresponding reduced set, its linear projection scatterplot shows the most variance directions in the leaf region. As expected with PCA, amplified by the fact that it focuses on a region of the whole data, this often shows insights about the intrinsic local structure of the data. In particular, we can usually identify the coordinate axes (PCs) with meaningful quantities or degrees of freedom (labeled in the scatterplot below each leaf in fig. 5). For example, leaves L1, L2, L4, L7, L14, L15 and L16 contain objects of a specific type, such as L1 containing high-heel and high-ankle boots and sandals, or even isolating a class, such as L14 isolating bags. Interestingly, we find all these leaves often detect similar degrees of freedom: usually, PC1 is (overall) pixel intensity in the image and PC2 is object size (either height or width). Again, this is the result of minimizing reconstruction error, which projects along maximum-variance directions. Also, since overall intensity and object size are correlated, this is seen in the PCA plots as triangular shapes whose vertex corresponds to low-intensity images (caused by either overall low intensity or by a very small object). This can be clearly seen in leaf L4 (consisting of T-shirts and tops), as well as in other leaves: L7/L15 intensity-width; L1/L2/L16 intensity-height; L14 intensity-size of bags. The density of the projections in the 2D embedding is also informative. For example, in leaf L4, narrow tops are much more frequent than thin tops.

Some leaves find degrees of freedom not aligned with a PC direction, but visible in the scatterplot, such as elongated clusters. These often correspond to different classes, where the direction of elongation is some meaningful degree of freedom within the cluster. For example, as shown in the bottom of fig. 5, leaf L3 contains two types of clothes: "shirts" and "shoes". PC1 represents the class discrimination direction, while PC2 represents intensity. Local inspection of each cluster separately demonstrates that both clusters have maximum variation indicating pixel intensity. The shoe cluster also has variations in "weight" (lightweight shoes below and chunky shoes above). The "shirts" cluster has variation in T-shirt width. Leaves L4, L15, and L16 have almost a single class: shirts, dresses, and slim sneakers, respectively, and all of them have similar variations, i.e., intensity as PC1 and width or thickness as PC2. Finally, note that the appearance of clusters or complex structure within a leaf scatterplot suggests it would be worth either further expanding that leaf or using a deeper tree, which could be done in an interactive way.

This dataset provides evidence of what patterns are real and whether they are detected by a given DR method. If clusters appear in a PCA plot, they are real (unlike in a $t$-SNE or UMAP embedding). This is because, in an orthogonal linear projection, distances either decrease or do not change, but cannot increase. Another example is the fact that the image overall intensity arises throughout the PCA tree leaves as a principal component. This is not surprising: intensity changes in this dataset are large, and thus carry considerable variance, which the local PCAs pick. Even the global PCA is able to pick that too and locate low-intensity images in the lower-left area of the 2D embedding (see appendix fig. 10). However, both $t$-SNE and UMAP do not; we have verified that low-intensity images are mapped all over the embedding in a haphazard way. A similar effect happens with the Amazon reviews document dataset (appendix section E): individual document classes appear in PCA plots as "streaks", caused by the document length (which changes the norm of the TF-IDF feature vector), and converging on the area occupied by short documents. Again, such an important pattern is lost in the $t$-SNE and UMAP embeddings.

## 7 Conclusion

We have defined a new model for data visualization, the PCA tree, as a tree autoencoder that produces a set of hierarchical low-dimensional maps, and have given an algorithm to learn it by minimizing the reconstruction error, based on Tree Alternating Optimization (TAO). PCA trees are fast for training and out-of-sample inference, and highly interpretable. We hope they will provide a useful tool for data visualization. Beyond this, tree autoencoders have application in dimensionality reduction in general, as well as in clustering, subspace clustering, data compression, fast search and other problems that we are exploring.

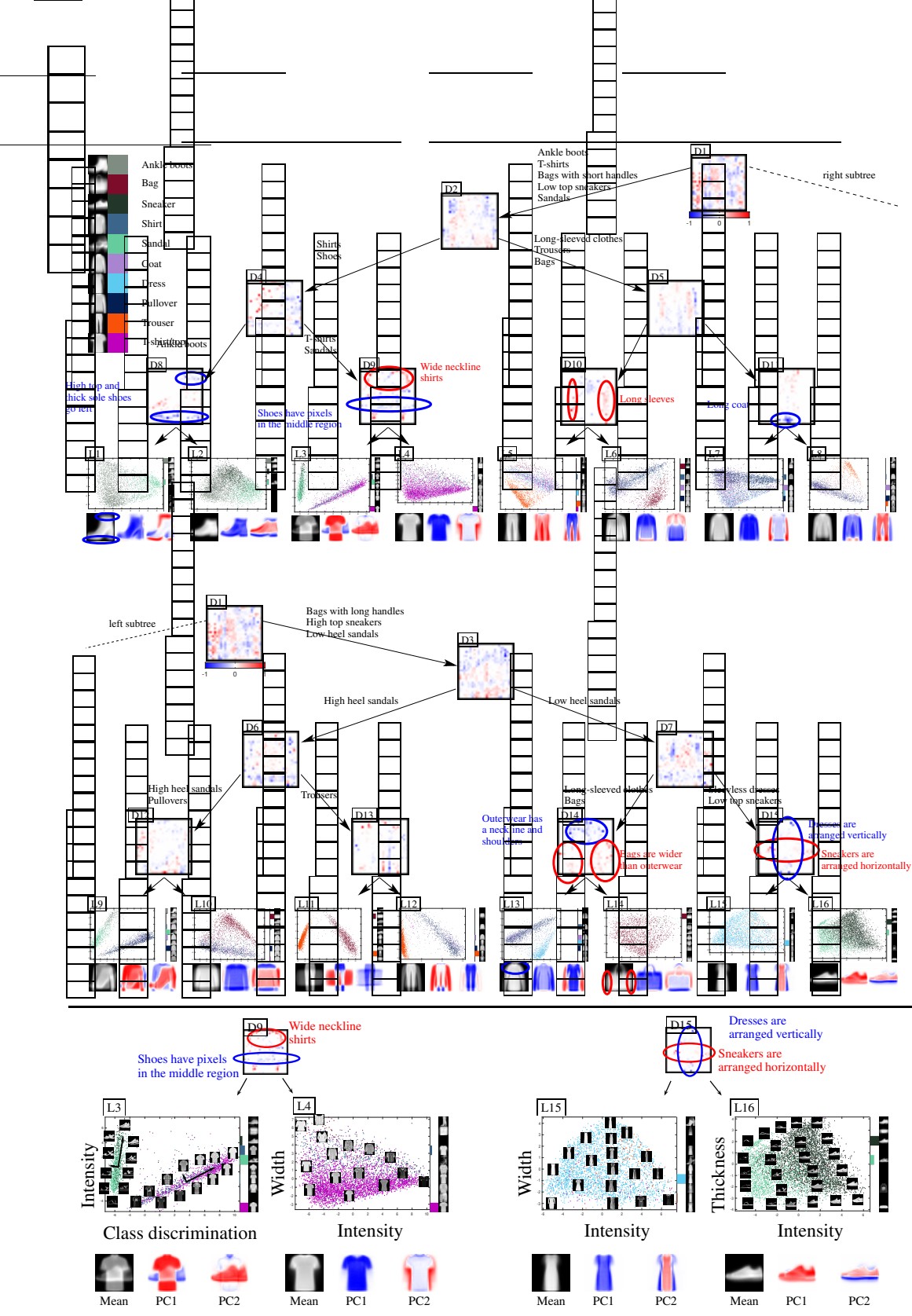

Figure 5: PCA tree trained on the Fashion MNIST dataset ($\Delta = 4$, $\lambda = 100$, RMSE per pixel and image of 0.158). The decision nodes' weight vectors (and the leaves' PCs $\mathbf{U}_j$) are shown as $28 \times 28$ images, with negative/zero/positive values colored blue/white/red, respectively. Each leaf shows a 2D PCA scatterplot of its RS (instances reaching it), and below it the mean $\boldsymbol{\mu}_j$ as a grayscale image and the 2 PCs $\mathbf{U}_j$ as color images. To the right of the scatterplot, a bar chart displays class proportions and class means. The legend (top left) shows, for each class, its mean (grayscale image), color (for the scatterplots) and description. For visualization purposes, the tree is split into its left and right root subtrees (fig. 8 shows the whole tree). The bottom panel zooms into two regions of the tree, for decision nodes 8 and 14. You may want to zoom into the figure to see more details.

## Acknowledgments and Disclosure of Funding

Work supported by NSF award IIS–2007147. We thank Magzhan Gabidolla for help with the TAO implementation.

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

# A   Appendix / supplemental material

## A.1   Algorithm pseudocode

**input** training set $\{\mathbf{x}_n\}_{n=1}^N$;
      sparsity hyperparameter $\lambda \geq 0$;
      initial tree $\mathbf{T}(\cdot; \boldsymbol{\Theta})$ of depth $\Delta$ with parameters $\boldsymbol{\Theta} = \{\mathbf{w}_i, w_{i0}\}_{i \in \mathcal{D}} \cup \{\mathbf{U}_j, \boldsymbol{\mu}_j\}_{j \in \mathcal{L}}$
$\mathcal{N}_0, \ldots, \mathcal{N}_\Delta \leftarrow$ nodes at depth $0, \ldots, \Delta$, respectively
for each $i \in \mathcal{N}$: generate $\mathcal{R}_i$ (instances that reach node $i$) on tree $\mathbf{T}$
**repeat**
  **for** $d = \Delta$ **down to** $0$
    **parfor** $i \in \mathcal{N}_d$
      **if** $i \in \mathcal{L}$ **then**
        $\{\mathbf{U}_i, \boldsymbol{\mu}_i\} \leftarrow$ PCA with $L$ components on $\mathcal{R}_i$
      **else**
        generate pseudolabels $\overline{y}_n$ and weights $\overline{w}_n$ for each instance $\mathbf{x}_n \in \mathcal{R}_i$
        $\{\mathbf{w}_i, w_{i0}\} \leftarrow$ fit $\ell_1$-regularized weighted binary classifier on $\{(\mathbf{x}_n, \overline{w}_n, \overline{y}_n)\}_{n \in \mathcal{R}_i}$
               with penalty $\lambda$
      **end if**
    **end for**
  **end for**
  for each $i \in \mathcal{N}$: update $\mathcal{R}_i$ on tree $\mathbf{T}$
**until** stop
**return** $\mathbf{T}$

Figure 6: Pseudocode for the PCA tree optimization algorithm.

# B  Experimental Details on Datasets and Algorithms

**MNIST and Fashion MNIST**  We use only the training set of size ($N = 60000$) and scale pixel values to $[0, 1]$ by dividing by 255.

**20newsgroups**  We use only the training set ($N = 11314$) without headers, footers, and quotes. The text is converted to lowercase, and all URLs, email addresses, punctuation, and numbers are removed. Stopwords are removed using NLTK [3], and the dataset is lemmatized using SpaCy [18]. We then apply scikit-learn's `TfIdfVectorizer` with default parameters, resulting in a dimension of $D = 72764$.

**Letter**  Recognition task for classifying 26 capital letters in the English alphabet [25]. We use only the training set ($N = 16000$). Standardization is applied to all features.

**Amazon Reviews**  We subsample text reviews from the large collection in [15] (version from 2014). Reviews are selected from the following categories:

- **Toys and Games**: [Learning & Education: (Mathematics & Counting, Reading & Writing), Games: (Board Games, Card Games)]
- **Beauty**: [Skin Care: (Hands & Nails, Face), Hair Care: (Hair Color, Shampoo)]

We use scikit-learn's `CountVectorizer` to extract unigrams from the raw texts. We set `stop_words='english'`, `min_df=0.0002` and `max_df=0.2`. Then, we apply the `TfidfTransformer`. The resulting dataset size is $N = 20000$ and $D = 9430$.

The summary of the datasets is the following:

|     | MNIST | FMNIST | Letter | 20newsgroups | Amazon reviews |
|-----|-------|--------|--------|--------------|----------------|
| $N$ | 60000 | 60000  | 16000  | 11314        | 20000          |
| $D$ | 784   | 784    | 16     | 72764        | 9430           |
| $K$ | 10    | 10     | 26     | 20           | 8              |

**Implementation**  We implement PCA tree using a combination of C++ and Python. Decision node optimization is implemented in C++ using LIBLINEAR [10], and leaf optimization is implemented in Python using PCA from the scikit-learn library with `svd_solver=arpack` and `n_components=2`. We use the following hyperparameters: $\lambda = 10$ for MNIST and Fashion MNIST, $\lambda = 1$ for Letter, and $\lambda = 0.01$ for 20newsgroups and Amazon Reviews. The algorithm includes an early stopping criterion that terminates training if there is no decrease in the training error for 3 iterations with a change of less than $10^{-3}$.

For $t$-SNE, we use the scikit-learn implementation (the default number of iterations is $1\,000$). For UMAP, we use the implementation from `https://umap-learn.readthedocs.io/en/latest/` (the default number of iterations is 500 if the dataset size has fewer than $10\,000$ points and 200 otherwise).

**Hardware**  All experiments are conudcted on a Intel(R) Xeon(R) CPU E5-2699 v3 @ 2.30GHz, 256GB RAM.

# C   Additional experimental results

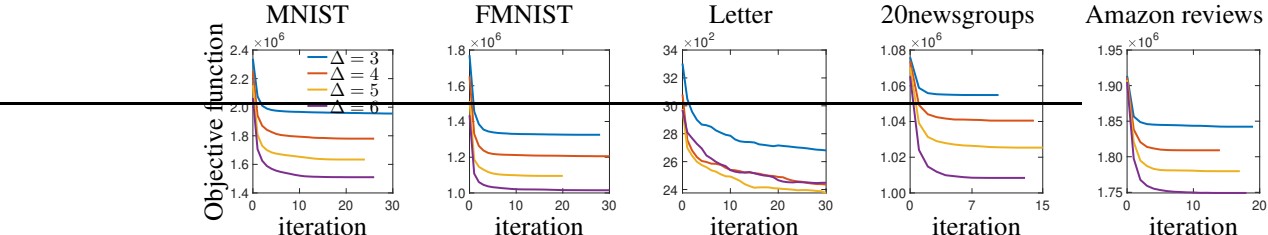

Figure 7: Objective function across training iterations for different datasets.

# D Interpretation and comparison of PCA tree visualizations

## D.1 Fashion MNIST

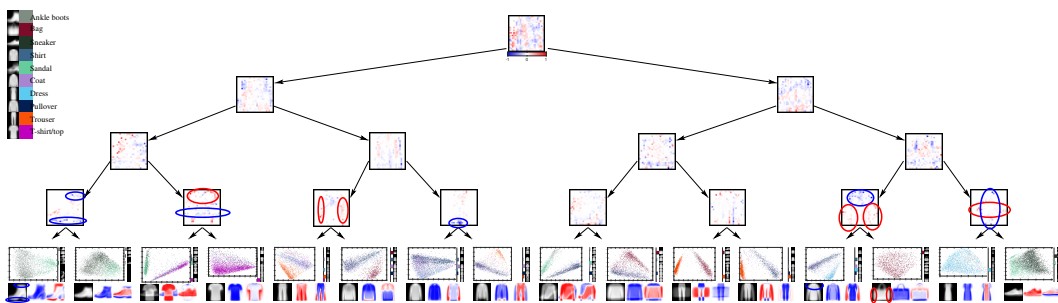

Figure 8: The same tree as in Figure 5 in unsplitted view.

The tree structure allows visualization of PCA scatterplots not only for the leaves but also for the decision nodes, as shown in Fig. 9. Note how PCA projections of each class or group of classes separate distinctly from each other.

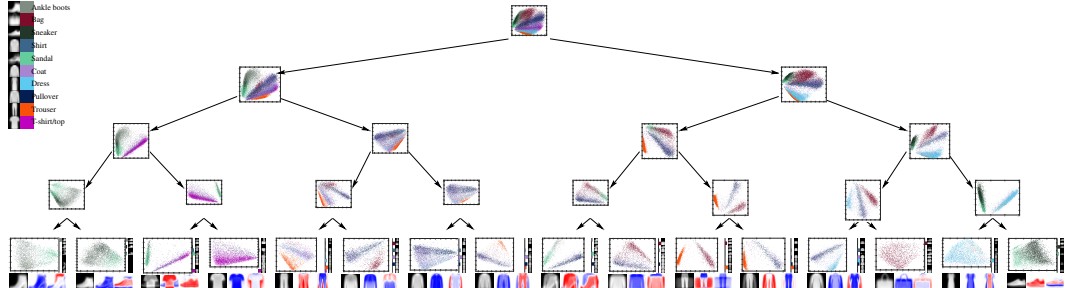

Figure 9: The same tree as in fig. 5, but with decision nodes representing PCA projections on their respective reduced sets.

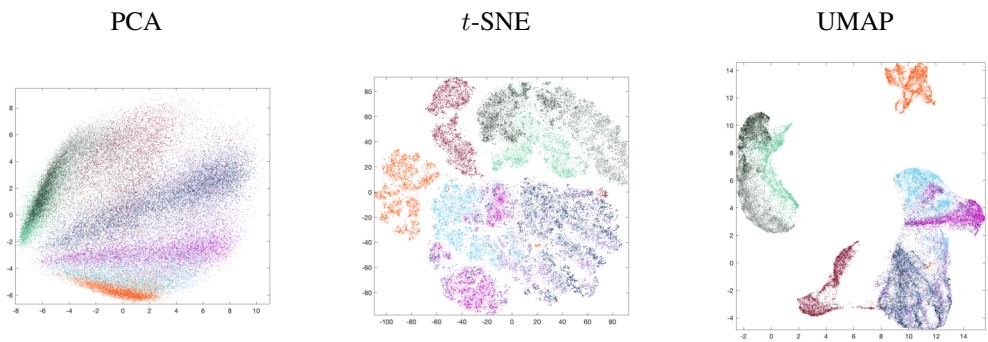

Figure 10: Scatterplots of different dimensionality reduction methods for the same dataset as in fig. 5 (Fashion MNIST) ($N = 60000$, $D = 784$). Class colors match the labels from fig. 5

### D.2 MNIST dataset

We evaluate our algorithm on the MNIST dataset. It is noticeable that almost half of the PCA tree leaves are nearly pure, containing mostly samples from a single class, while the other half contain classes that can be linearly transformed into each other. For example, leaf L16 contains three classes: 4, 9, and 7. By adding the top bar and removing the middle bar, we can achieve the following linear transformation: $4 \rightarrow 9 \rightarrow 7$, as shown by the first principal component. The second component represents pixel intensity, so digits go from thin to thick from bottom to top. Now, consider the pure leaf L12. The digit 1 changes its inclination from "to the right" to "to the left" on the x-axis and its intensity from bottom to top.

PCA overlaps nearly all data points except for those representing the digit "1".

t-SNE distorts the global structure of the data. It forms clusters such that each cluster represents a separate class, which can be misleading in terms of distances. For example, the cluster for the digit "2" appears close to the cluster for "1" and far from "3", but in the high-dimensional space, the relationship is the opposite. A similar issue occurs with the digits "0" and "5".

UMAP also distorts the positions of data points. It may show that the digit "3" is closer to "6" than to "4", while in the original high-dimensional space, the relationship is the opposite.

Such distortions can result in wrong interpretations of the data's structure and cluster relationships.

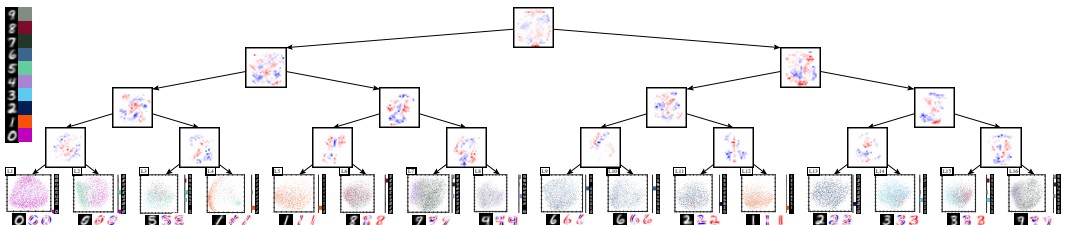

Figure 11: Trained PCA tree structure on the MNIST dataset with a reconstruction loss of 0.29

PCA        t-SNE        UMAP

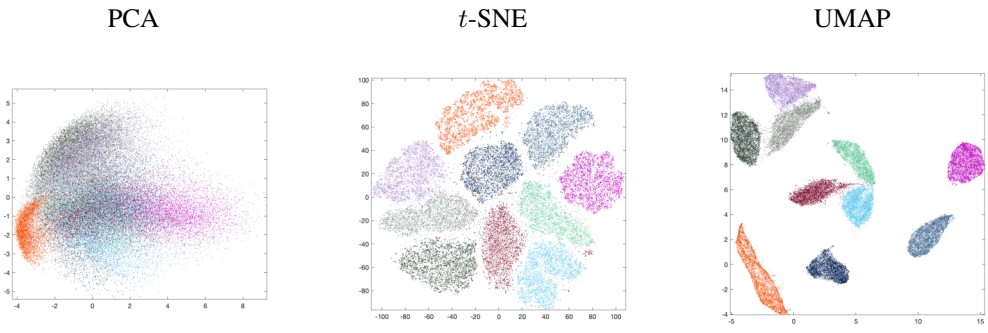

Figure 12: Scatterplots of different dimensionality reduction methods for the same dataset as in fig 11 ($N = 60000$, $D = 784$). The class colors correspond to the labels in fig. 11

# E    Amazon Reviews Dataset

Consider now the dataset from a different domain, specifically text data. We use the Amazon Reviews dataset, represented as TF-IDF features, containing 20,000 samples and 9,430 features. The decision tree structure, illustrated in fig. 13) allows us to visualize the top words corresponding to the highest and lowest weight values, indicating important words at each decision node. Furthermore, using highest and lowest entries for both principal components, can be demonstrated variation across reviews, see table 1

By visually inspecting the leaves, we can identify similar reviews from different classes. For example, the first leaf contains many samples from different classes, but most of these samples are short reviews about "returns" and "refunds". Elongated clusters within the leaf reveal reviews about different topics, such as games (left cluster) and cosmetics (right cluster). Additionally, reviews from different classes may pertain to items with similar purposes. In leaf L3, there are four classes, but all are related to games: the top cluster is about "funny learning", the left cluster is about "teaching children to count using cards", and the right cluster is about "card games".

Similar to previous examples, this tree has principal components that represent semantic or class separation. For instance, in leaf L5, the first principal component separates children's games from adult games, while in leaf L11, the first principal component distinguishes one class from the rest. Note that some leaves are pure, such as leaf L7 or L10, and within these leaves, clusters correspond to topics. Leaf L7, for example, is split into three topics: "cuticles" (left), "polishing" (top), and "nail growing" (right), although all samples are from the same class.

$t$-SNE and UMAP significantly overlap most of the classes. While PCA projections present some cluster elongations, they also show overlapping classes and do not reveal the inner local structure.

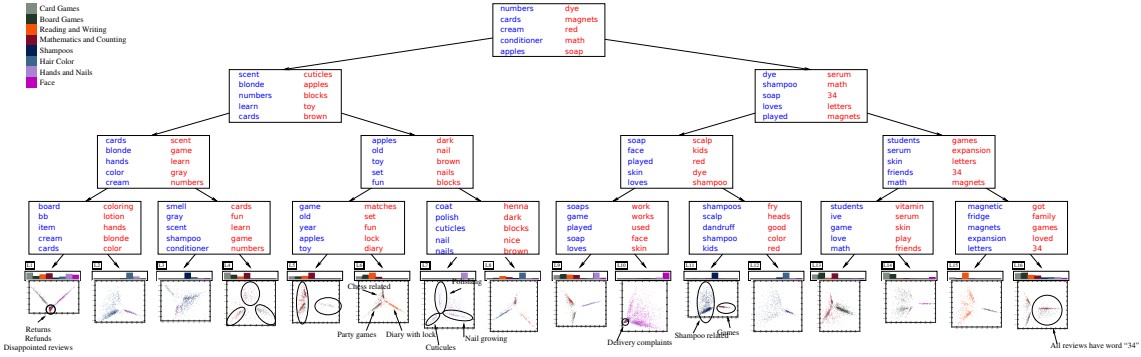

Figure 13: Trained PCA tree structure on the Amazon reviews dataset with a reconstruction loss of 0.9004

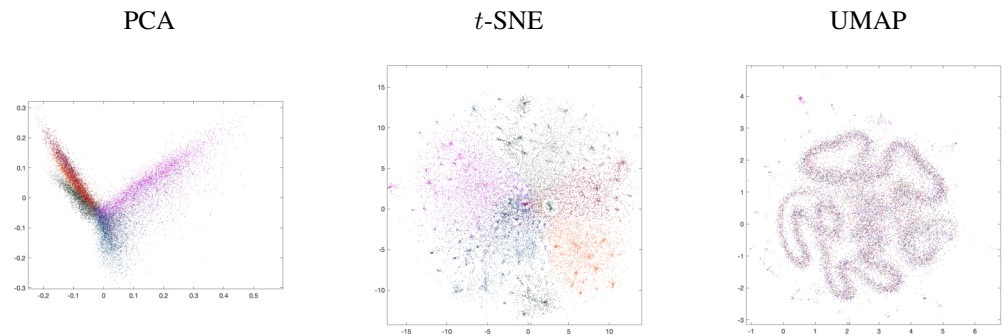

Figure 14: Scatterplots of different dimensionality reduction methods for the same dataset as in fig. 13 ($N = 20000$, $D = 9430$). The class colors correspond to the labels in fig. 13.

| Leaf | 1st principal component | | 2nd principal component | |
|------|------|------|------|------|
| L1 | cream, hand, skin, hands, bb | cards, deck, card, playing, box | cards, cream, hand, skin, hands | item, expected, book, dies, magnetic |
| L2 | hands, hand, dry, lotion, greasy | color, blonde, brown, natural, coloring | blonde, light, ash, highlights, toner | color, hands, lasts, lotion, hand |
| L3 | conditioner, shampoo, using, clean, soft | gray, scent, color, cover, coverage | gray, conditioner, color, cover, coverage | scent, clean, strong, suave, nice |
| L4 | cards, game, card, play, players | numbers, letters, learn, learning, loves | learn, easy, fun, children, help | numbers, cards, card, number, letters |
| L5 | apples, game, friends, adult, version | year, old, toy, loves, bought | toy, learning, child, toys, educational | year, old, game, apples, play |
| L6 | diary, lock, key, journal, daughter | fun, game, family, playing, play | set, chess, pieces, board, nice | fun, diary, lock, game, key |
| L7 | nails, grow, stronger, coat, weak | cuticles, cuticle, oil, nail, dry | nail, polish, coat, base, art | cuticles, nails, cuticle, oil, dry |
| L8 | blocks, nice, quality, wood, usa | brown, color, dark, medium, black | nice, board, pieces, price, smells | blocks, brown, color, dark, medium |
| L9 | soap, hand, hands, soaps, scent | loves, son, played, game, daughter | loves, soap, son, daughter, absolutely | played, game, fun, family, blast |
| L10 | skin, face, sensitive, dry, acne | price, buy, bought, good, amazon | face, wash, clean, cleanser, acne | skin, price, work, sensitive, dry |
| L11 | kids, game, love, playing, fun | shampoo, dandruff, scalp, shampoos, clean | dandruff, scalp, clear, anti, shoulders | shampoo, kids, love, suave, volume |
| L12 | good, smell, smells, works, really | red, color, dye, brown, dark | dye, purple, bleach, black, did | red, good, smell, smells, brown |
| L13 | game, play, family, fun, playing | math, love, students, facts, learning | love, students, perfect, price, absolutely | math, game, fun, skills, kids |
| L14 | serum, skin, vitamin, acid, hyaluronic | game, play, friends, fun, family | friends, played, playing, family, game | play, kids, easy, old, time |
| L15 | expansion, game, packs, cards, pack | letters, magnets, fridge, magnet, magnetic | magnets, fridge, hold, strong, fall | letters, expansion, game, case, lower |
| L16 | 34, shampoo, color, does, doesn | loved, games, game, family, board | loved, gift, birthday, niece, journal | games, game, board, fun, family |

Table 1: Two principal components of each leaf in PCA tree 13, showing the words corresponding to the top 5 positive and negative entries of the $U_j$ columns (principal components' coefficients).

