# OpenReview forum: "The tree autoencoder model, with application to hierarchical data visualization"
_NeurIPS.cc/2024/Conference — NeurIPS 2024 poster_

### Official Review · Reviewer_Aeto · 2024-07-08

**Soundness:** 3
**Presentation:** 4
**Contribution:** 3
**Rating:** 7
**Confidence:** 3

**Summary:**

This paper proposes a dimensionality reduction approach that comprises a tree of local PCA projections. The authors make the following contributions:
1. A mathematical description of their approach.
2. An optimisation algorithm and pseudocode to fit their method to a training set, together with the computational complexity of inference and training of the model giving evidence of the excellent properties of the approach.
3. Experiments with benchmark datasets of various types confirming the claims around scalability and computational complexity, the reconstruction error, and examples of using the approach to explore and interpret datasets.

**Strengths:**

1. This is an interesting, high quality, well written and argued paper that addresses a significant problem in data exploration and visualisation. Whilst there is a plethora of approaches for dimensionality reduction of complex datasets, no one approach has really emerged. The proposed approach marries the benefits of PCA (including consistent distances) with a hierarchy which allows understanding of sub-regions of the space, giving a representation that is explainable and fast to compute compared to other approaches such as UMAP or t-SNE.
2. The related work is to the point and identifies the state of the art / popular dimensionality reduction methods. They highlight the novelty of their work against this background and why their approach is desirable. The work is original to my understanding, building on hierarchical DR methods, particularly hierarchies of sparse oblique trees.
3. The mathematical definition of their approach is clear and concise and the proofs around the optimisation clearly highlight why their approach has good properties around the possibility of parallelism, and benefits in computational complexity. In particular, the discussion around the surprising properties in the computational complexity was very interesting.
4. There is a nicely articulated discussion of experimental results, which confirm the properties around the optimisation algorithm, its scalability and reconstruction properties for a well known set of benchmark datasets of various types: image and text, as well as a detailed description of examples of the local structure found in the Fashion MNIST dataset. The experiments generally confirm the claims stated in the introduction.
5. Whilst there is no code available at this stage, the benchmark datasets are readily obtainable and there is sufficient detail around the experiments that the work could be reproduced with some effort.

**Weaknesses:**

1. There are several hyperparameters for the approach, most notably $\lambda$ and $\Delta$. However, the paper does not describe how to set these parameters nor the sensitivity of the approach to different values.
2. At the moment the approach builds out a full tree to depth $\Delta$, but it would be better if the tree could be deeper in particular areas depending on the data in the associated region. In fact, several nodes would be improved in the example in fig. 4 if the tree were deeper - e.g. L3, L5, L8 and L11.
3. I couldn't see the parameter settings for the experiments with t-SNE and UMAP.

**Questions:**

How to set the hyperparameters? What is the sensitivity to hyperparameter values? How significant is the effect of local minima to this optimization problem?

Graph-based similarity approaches have several advantages and disadvantages compared to distance-based methods like the approach in this paper. Given that the proposed approach sub-divides data space to explore local structure, is there an opportunity to unify the two approaches in the leaves?

**Limitations:**

The paper describes limitations of the proposed approach, although to some extent these are limitations for all dimensionality reduction / visualisation methods. The paper does not speak much to the effect of local minima. The paper does not explore the sensitivity to hyperparameters. It might also be interesting to explore dimensionality reduction in synthetic datasets where there is known structure, e.g., the noisy swiss roll datasets, box datasets and others, for example used in Lee and Verleysen, Nonlinear Dimensionality Reduction (2007) as an approach to compare a wide set of methods.

---

> ### Author Rebuttal · Authors · 2024-08-07
>
> Thanks for your very thorough review.
>
> - *Hyperparameter setting and sensitivity.*
>
>   In principle, since an autoencoder defines a supervised problem, the hyperparameters might be chosen using a hold-out validation set. However, for visualization purposes we set them manually as described next.
>
>   The tree depth $\Delta$ controls the granularity of the result and the level of detail seen. The deeper the tree, the lower the reconstruction error (eventually reaching zero error), but the more leaves it has and the harder it is to interpret. Larger datasets may require deeper trees. For visualization purposes, we tried depths up to 4 (= 16 leaves).
>
>   The hyperparameter $\lambda$ controls the decision node hyperplane sparsity. The larger $\lambda$ is, the fewer nonzero weights we have (and, if all weights in a hyperplane are zero, that node can be pruned, as it becomes redundant). In our visualization experiments we used relatively few nodes, so we set $\lambda$ to a value (100, in line 273) that results in zeroing unnecessary weights, which improves interpretability, but is not aggressive enough that nodes are actually pruned. Empirically, we observe $\lambda$ has to vary exponentially (say, from $10^{-3}$ to $10^6$) for it to affect significantly the results.
>
>   We'll note this in the paper.
>
> - *It would be better if the tree could be deeper in particular areas.*
>
>   Yes, in fact we mentioned this briefly in lines 319ff. This could be done in different ways. One is by tuning the hyperparameters $\lambda$ and $\Delta$ (by cross-validation or interactively); note that using a deep enough tree with high enough sparsity will prune nodes and result automatically in an irregular tree structure. Another way, not explored here, is to have the training algorithm expand nodes on the fly in some way. Perhaps the simplest way (noted in line 321) is to have the user expand nodes selectively in an interactive way (and running the algorithm to re-optimize the new tree).
>
> - *Parameter settings for the experiments with t-SNE and UMAP.*
>
>   We set the hyperparameters for t-SNE and UMAP to their default values in the implementations we used (perplexity 30 and 15, respectively).
>
> - *Effect of local minima.*
>
>   Given the nonconvexity of the problem, this is an unavoidable issue (as is the case with t-SNE, for example). At present, we simply try different random initializations of the decision node hyperplane directions (the hyperplane bias is set to achieve a 50%/50% partition of the samples, so all leaves receive about the same number of samples in the first iteration). We have explored other ideas, for example doing a bottom-up merging as in agglomerative clustering, but we cannot recommend them as consistently better.
>
> - *Combining graph-based similarity approaches with subdividing the data space.*
>
>   We have some ideas about that which hopefully we'll develop at some point...

---

> > ### Comment · Reviewer_Aeto · 2024-08-12
> >
> > Thank you for your response to the questions in my review and the clarification of parameter settings. What you say is all reasonable. After considering the response, I will maintain my current scoring.

---

### Official Review · Reviewer_DywJ · 2024-07-12

**Soundness:** 3
**Presentation:** 3
**Contribution:** 2
**Rating:** 5
**Confidence:** 3

**Summary:**

This study proposes PCA trees as a hierarchical data visualization tool. PCA trees hierarchically partition the data space using hyperplanes, with the partition pattern represented by a binary tree. PCA is then applied to each leaf node of this binary tree. The parameters of a PCA tree consist of coefficients defining the hyperplanes for splitting at internal nodes and projection matrices for PCA at leaf nodes. The function representing the fitting of the PCA tree to the data satisfies a separability condition with respect to its parameters. Therefore, we can perform effective alternating optimization. The characteristics of PCA trees are also verified through visualization of real-world datasets such as Fashion MNIST.

**Strengths:**

* Formulation that reduces hierarchical data visualization to the optimization of a certain type of decision tree.
* Proposal of an algorithm for efficient execution of this optimization.
* Thorough explanation of the results of applying the proposed method to real-world datasets such as Fashion MNIST.

**Weaknesses:**

PCA trees divide the data space into multiple disjoint subsets and project the data points in each subset into a different low-dimensional space. Therefore, it is debatable whether this truly achieves dimensionality reduction in the strict sense. The description of ''dimensionality reduction'' in the abstract and introduction may be misleading. It might be more appropriate to view this as a data visualization technique. Furthermore, there are doubts about whether the visualization provided by PCA trees is intuitively easy to understand. Analysts need to conceptualize and integrate complex hierarchical partitions of high-dimensional space along with projections in various directions. In particular, intuitively grasping the positional relationships of data points between different nodes might be challenging. However, given that the assessment of visualization quality is largely subjective, PCA trees may still have value when used complementarily with other visualization techniques.

**Questions:**

What strategies or techniques are employed to avoid converging to local optima?

**Limitations:**

The paper acknowledges that the solution may converge to local optima. Further discussion on strategies to mitigate this issue would enhance the work.

---

> ### Author Rebuttal · Authors · 2024-08-07
>
> Thanks for your review.
>
> - *Does our PCA tree truly achieve dimensionality reduction in the strict sense?*
>
>   It does indeed, but we need to define the latent space and its coordinates accordingly. This latent "space" consists of 1) the discrete index of a leaf (local latent space) and 2) a continuous latent vector (within that local latent space). For example, in figure 4 (Fashion MNIST dataset), an image is "projected" to a triplet $(j,z_1,z_2)$ (index and two real numbers), and reconstructed back from such a triplet. See lines 107ff. The encoding and decoding mappings are correspondingly defined based on that space. This makes our autoencoding framework well defined mathematically, and indeed the right way to handle multiple latent spaces. We do acknowledge that it is unconventional within traditional DR, where autoencoders are traditionally seen as purely continuous and on a single latent space.
>
> - *Is the visualization provided by PCA trees intuitively easy to understand?*
>
>   Using multiple latent spaces is not a perfect solution. We acknowledge our work's limitations in section 1 and caution practitioners to use multiple methods. But, are methods using a single space, such as t-SNE, really "easy to understand"? The necessary, misleading distortions arising from using a single latent space have been well documented. Ultimately, all methods must make compromises to arrange high-dimensional data onto some kind of visualization. For a visualization to be useful, it should extract meaningful patterns from the data, and we think that hierarchical local DR has important, complementary benefits in this respect compared to global-space methods (see section 1).
>
> - *Strategies or techniques are employed to avoid converging to local optima.*
>
>   Given the nonconvexity of the problem, this is an unavoidable issue (as is the case with t-SNE, for example). At present, we simply try different random initializations of the decision node hyperplane directions (the hyperplane bias is set to achieve a 50%/50% partition of the samples, so all leaves receive about the same number of samples in the first iteration). We have explored other ideas, for example doing a bottom-up merging as in agglomerative clustering, but we cannot recommend them as consistently better.

---

> > ### Comment · Reviewer_DywJ · 2024-08-10
> >
> > Thank you for your response.
> >
> > Regarding the initialization method to avoid local optima, while it doesn't seem particularly novel, I think it's a reasonable approach. If the paper is accepted, I believe this should be included in the final manuscript. However, I don't think it's significant enough to improve the overall evaluation of the paper.
> >
> > Concerning the points I raised as Weaknesses, I understand your perspective, and I agree that there is some merit to what you're saying. However, this aspect inevitably involves subjectivity, making it difficult to draw a clear conclusion. That's precisely why I included these points under Weaknesses and not under Questions. As I mentioned in my initial review, I believe PCA trees may have value when used to complement other visualization techniques. Your definition of the latent space dimension was helpful in clarifying the discussion.
> >
> > After considering all aspects, I've decided to maintain my original evaluation.

---

### Official Review · Reviewer_2dBg · 2024-07-12

**Soundness:** 2
**Presentation:** 2
**Contribution:** 1
**Rating:** 4
**Confidence:** 4

**Summary:**

The paper proposes a hierarchical data visualisation method that is a combined use of oblique decision tree and PCA over leaf nodes. In experiments, the authors demonstrate the visualisation results of the proposed method for a few datasets and compare the reconstruction errors with standard PCA.

**Strengths:**

The demonstrated visualisation results are useful.

**Weaknesses:**

Very limited originality. The proposed method works by firstly training an oblique decision tree and then applying PCA over examples assigned to each leaf nodes. This is a straightforward use of very classical existing methods. It hardly qualifies as a new method…

Since this paper contributes to hierarchical data visualisation, their experiments should focus on comparison with the use of existing hierarchical DR for data visualisation.

Most theoretical results/analysis on PCA tree come from simple applications of classical PCA results. The three theorems do not seem to present any significant result, for which I invite the authors to comment on.

The method description is quite hard to follow, which appears like a piece of overly complicated writing for simple method.

**Questions:**

How is the oblique tree trained in their experiments?

I invite the authors to comment on why the results from the three theorems are important.

**Limitations:**

No discussion on limitation.

---

> ### Author Rebuttal · Authors · 2024-08-07
>
> We thank you for your questions, but it seems there are several misunderstandings in your review. Hopefully our answers below will clarify them.
>
> - *Very limited originality [...] The method description is quite hard to follow, which appears like a piece of overly complicated writing for simple method [...] How is the oblique tree trained in their experiments?*
>
>   It appears you may have misunderstood our algorithm. If, as you say, our algorithm consisted of 1) training (somehow) an oblique decision tree and 2) given this tree, applying PCA over the examples assigned to each leaf, then we'd agree that this is straighforward (and suboptimal). But what we do is different, as follows. 1) We define a global objective function (eq. (1)), the reconstruction error, over all the tree parameters (at the decision nodes, which route the examples to the leaves; and at the leaves, which effect the DR). 2) We propose a new algorithm that monotonically decreases this objective function until convergence, thus iteratively rerouting examples to find an optimal, hierarchical partition of the feature space, and correspondingly optimal local dimensionality reduction at each leaf region. The algorithm is based on alternating optimization over the tree nodes' parameters. Overall, then, we define a new model (the tree autoencoder) and train it jointly over all its parameters (which is not straighforward due to the nondifferentiable, nonconvex nature of the tree).
>
> - *Comparison with the use of existing hierarchical DR for data visualisation.*
>
>   As noted in our related work, hierarchical DR is either soft (which makes it hard to interpret) or hard (for which, as far as we know, there is no work that optimizes some desirable objective function over the tree). For comparison purposes, we could have constructed a tree using a traditional algorithm such as CART (greedy top-down recursive partitioning) and fitted PCA at the leaves, but this works extremely poorly (it generates huge trees with a large error). Instead, we opted to compare with state-of-the-art methods such as t-SNE.
>
> - *The three theorems do not seem to present any significant result.*
>
>   Our theorems have several purposes. 1) They make it possible to apply alternating optimization over the decision nodes and leaves, which require the solution of certain precisely-defined optimization subproblems. 2) They ensure that our algorithm monotonically decreases the objective function. 3) They indicate under what conditions nodes can be trained in parallel. 4) For the leaves, the Reduced Problem theorem gives an optimality condition akin to a KKT condition for differentiable problems.
>
> - *How is the oblique tree trained in their experiments?*
>
>   We do not just train an oblique tree. We train a tree autoencoder having oblique decision nodes and linear autoencoders at the leaves, jointly over all the parameters. The fundamental technique is alternating optimization over the nodes, whose solution is given by our theorems. Section 4 gives the details and figure 5 the pseudocode.
>
> - *No discussion on limitation.*
>
>   We do have a discussion on limitations, which starts in line 58.

---

> ### Comment · Reviewer_2dBg · 2024-08-11
>
> Thank you for clarifying the method and providing other explanations.  I  misunderstood the part that you jointly learn the tree (decision nodes) and the projection parameters.
>
> Regarding to comparison with other hierarchical methods, the point is to show the difference between the visual patterns captured by those methods and yours, even the compared ones are “poor” or work by completely different ways from yours.  Intuitively, there are straightforward ways of producing a hierarchical visualization result, in a similar fashion to those in your Fig 4.  For instance, you can run a hierarchal clustering algorithm and then visualise each cluster. Also, you can obtain hierarchy by grouping feature dimensions. Another way is by what I thought you did, which is to train an oblique decision tree first and then run PCA.  There are works pursuing hierarchical visualisation aspects from different application domains, for instance,
>
> *  https://www.sciencedirect.com/science/article/abs/pii/S2214579619302102?fr=RR-2&ref=pdf_download&rr=8b17b9a228b7776b
>
> *  https://www.nature.com/articles/s41587-021-01186-x
>
> I think more research effort on investigating whether your proposed approach could reveal more interesting or under explored data patterns than naïve or existing approaches is needed.
>
> Given that I under-estimate the novelty but still don’t find it highly novel,  and that I find it lacks experiments on demonstrating advantage over other ways to produce a hierarchical data visualization, I will increase my score to 4, but I am reluctant to give a higher score.

---

> > ### Author Response · Authors · 2024-08-13
> >
> > Following your suggestion, we constructed some reasonable hierarchical DR baselines to compare with in the Fashion MNIST dataset (see PDF file). In all cases, we first construct a tree, then fix it and train PCA on each leaf (i.e., on the training points of each leaf). The tree is learned as follows:
> > - Via hierarchical clustering.
> > - As a "random median oblique tree", i.e., we construct a complete oblique tree of given depth where each decision node's hyperplane has a random direction and the bias selected to partition its dataset 50%/50%. This is actually how we initialize the tree in our algorithm.
> > - As a $k$-means recursive bipartition, i.e., we construct a complete oblique tree of given depth where each decision node's hyperplane is obtained by running $k$-means with $K=2$ clusters.
> > - We also tested the case $L=0$ of our PCA tree, i.e., to train a tree autoencoder of latent dimension zero, then fit PCA to the leaves (with $L=0$). This (not shown in the PDF file) is guaranteed to perform worse than the PCA tree, obviously. It performs similarly to the $k$-means recursive bipartition.
> >
> > As we noted in our earlier reply, all these approaches perform considerably worse than our PCA tree, in both reconstruction error and 2D scatterplots. In the PDF file, among other things, we report the reconstruction error for the points in the training set (which is an objective measure), for different numbers of leaves, and also show the PCA scatterplots in the leaves of the tree. This confirms the fundamental novelty and advantage of our approach: the ability to optimize not just the PCA mapping at the tree leaves, but crucially the tree itself. This adjusts the set of training points in each leaf and its PCA so the error is minimized. Learning the tree on its own and fitting PCA afterwards results in a much worse partition.
> >
> > The PDF is in the following anonymized link: https://anonymous.4open.science/api/repo/rebuttal-8298/file/rebuttal.pdf?v=f61e309a
> >
> > Below, we include the tabular and text parts of it (but not the scatterplots).
> >
> > | Number of leaves | PCA tree | agglomerative clustering | median tree | $k$-means bipartition |
> > | - | - | - | - | - |
> > | 8 | **22.07** | 36.22 | 30.80 | 24.19 |
> > | 16 | **19.57** | 29.87 | 28.72 | 21.43 |
> > | 128 | **15.40** | 23.98 | 23.49 | 16.38 |
> >
> > We compare our PCA tree with several two-stage approaches, where we first learn a tree and then apply PCA to each leaf. The PCA tree consistently shows significantly lower reconstruction error across varying numbers of leaves (Table 1).
> >
> > **Hierarchical Clustering**. We use agglomerative clustering to build a dendrogram and then apply PCA within each cluster. This method has notable drawbacks: it lacks an out-of-sample mapping, requires substantial computation time due to the pairwise distance calculations, and results in a poor tree (Fig. 1): 39379 samples (more than half of the whole dataset) end in a leaf, while many other leaves are nearly empty, containing only 1-2 points.
> >
> > **Median tree**. We split the data into two halves via a random-direction hyperplane at each decision node, resulting in a balanced partition at the leaves. This tree serves as initialization for our algorithm, so it is predictably worse.
> >
> > **$k$-means bipartition**. We construct a binary oblique tree by recursively running the $k$-means algorithm at each decision node, partitioning the data into two child nodes. The decision node hyperplanes (not shown) are not sparse, so they are harder to interpret than in our PCA tree. This works better and is able to identify some of the classes (since, after all, it is a form of clustering), but from a DR point of view it is not clear what meaningful patterns it reveals, particularly when it mixes multiple classes in the same leaf (see L7, L8, L11, L15, L16 in Fig. 2). It is important to note that in an unsupervised dataset, class labels are not available to color data points, making these scatterplots less insightful.

---

### Official Review · Reviewer_7zsa · 2024-07-17

**Soundness:** 3
**Presentation:** 4
**Contribution:** 2
**Rating:** 6
**Confidence:** 5

**Summary:**

The authors nicely describe an extension of [5] to unsupervised dimensionality reduction.
The paradigm is quite different from regular DR (unique scatterplot) as it yields a tree and thus multiple leafs.
There are some similarities with hierarchical clustering (to some extent).
The authors also emphasize a connection with autoencoders (though readers might understand NN autoencoders while here these are rather linear auto-encoder, equivalent to PCA).
Attention is dedicated to the computational complexity and parallelization of the tree structure.
There is one illustrative experiment in the paper and some other classical benchmarks are reported in the suppl.mat.

**Strengths:**

The paper is very nicely structured and written.
The method is intuitive and well explained.
There are sufficient experiments.

**Weaknesses:**

The novelty with respect to [5] might seem a bit limited.
The proposed visualization paradigm is unconventional and probably shares the same shortcomings as those of a dendrogram in hierarchical clustering.
The global structure is probably hard to tell, which is not different than the usual drawback of t-SNE and such (although a user study might answer that question).
To some extent (again), the proposed visualization is reminiscent of what was done in the past with SOMs (here the leafs would resemble a 1D SOM). The vast literature about SOMs might contain some variants with hierarchies or PCAs in the grid nodes... (open question)
Another possible comparison is with some older spectral DR techniques, working by stitching local 2D projections (back to a unique scatterplot then...).

**Questions:**

See above

**Limitations:**

The limitations appear quite clearly in the text, although the first limitation at the end of section 1 does not really make sense (t-SNE uses graph but is essentially DR and thus assumes/uses coordinates (or feature vectors)).

---

> ### Author Rebuttal · Authors · 2024-08-07
>
> Thanks for your review and in particular for the connections with other methods.
>
> - *Novelty with respect to [5].*
>
>   Since trees making hard decisions are not differentiable, we cannot use gradient-based methods and instead we use alternating optimization over the nodes. While [5] also used alternating optimization, there the model was a classification tree. Here we propose a different model, tree autoencoders, and adapt the alternating optimization algorithm accordingly. We believe this is the first time that this type of autoencoders have been proposed and successfully trained on high-dimensional data.
>
> - *The proposed visualization paradigm is unconventional.*
>
>   We agree with this, so we put significant effort in illustrating it in the experiments of section 6.3 and figure 4.
>
> - *Shortcomings as those of a dendrogram in hierarchical clustering.*
>
>   Hierarchical clustering algorithms (either bottom-up or top-down) do not optimize a global objective function over the tree and clustering. One critical advantage of our work is that it does define a global objective function of the parameters of the tree and the local DRs at the leaves, which our algorithm monotonically decreases at each iteration.
>
> - *The vast literature about SOMs might contain some variants with hierarchies or PCAs in the grid nodes.*
>
>   Thanks for noting SOMs, which we missed in our literature review. Indeed, we have now found some papers exploring hierarchical SOMs (all based on Rauber et al 2002). They are conceptually similar to what we describe in lines 73ff: one grows a hierarchical SOM recursively in a greedy way. This doesn't optimize a global objective function over all the tree and individual SOM parameters (indeed the regular SOM is well known not to optimize an objective function, its iterative process is stopped manually). The regular SOM also does not define an encoder and decoder explicitly, as it is essentially a vector quantization with a topology bias. We'll note this in the paper.
>
> - *Older spectral DR techniques that stitch local 2D projections to a unique scatterplot.*
>
>   We did mention this in the related work (lines 82-83). We did not compare with them in the experiments because we already included the single-scatterplot techniques that are currently most popular (PCA, t-SNE, UMAP).
>
> - *t-SNE uses graph but [...] thus assumes feature vectors.*
>
>   We can run t-SNE without feature vectors, as in MDS, if we have access to pairwise similarities between instances (or neighbor probabilities), although admittedly most applications do use feature vectors directly.

---

### Decision · Program_Chairs · 2024-09-25

**Decision:**

Accept (poster)

**Comment:**

The majority of the reviewers lean towards accept. I also find the proposed method interesting and novel, which may spark further research ideas. Thus I also recommend the paper be accepted.

In the camera-ready version, the authors should revise their paper based on the discussions with the reviewers. Additionally, for future work, I suggest the authors expand their comparisons to other hierarchical methods, such as those suggested by reviewer 2dBg. While the authors provided extra comparisons during the discussion phase, the use of reconstruction error is somewhat unsatisfying as the proposed method is explicitly designed to optimize this error. Other metrics should also be included in future work, although admittedly good metrics are hard to come by in these problems.